# Microbial drinking water quality deterioration during distribution and household usage, determined together with citizen scientists

**Paul W. J. J. van der Wielen** [1,2]*, **Stijn Brouwer** [1], **Marco Dignum** [3], **Merijn Schriks** [1]

**1** KWR Water Research Institute, Nieuwegein, The Netherlands, **2** Department of Microbiology, Faculty of Science, Radboud Institute of Biological and Environmental Science, Radboud University, Nijmegen, The Netherlands, **3** Waternet, Amsterdam, The Netherlands

* paul.van.der.wielen@kwrwater.nl

## Abstract

During transport and storage of drinking water the microbial water quality might deteriorate. Here, we studied the vulnerability of non-chlorinated drinking water produced by two treatment plants to deterioration, by involving citizen scientists. Citizen scientists in Amsterdam sampled their drinking water directly from their kitchen tap after overnight stagnation, after flushing and after storage in containers like reusable plastic bottles. Subsequently, prokaryotic cell counts, ATP concentrations and the prokaryotic community composition were determined in the laboratory. The results showed that citizen scientists were able to reliably sample drinking water. The microbiological parameters measured remained stable during drinking water transport in the distribution system, whereas overnight stagnation in the premises plumbing system could result in fluctuations in the microbial biomass parameters and slightly altered the prokaryotic composition. Drinking water storage in containers resulted in a substantially increase in microbial biomass, a decrease in bacterial diversity and a shift in bacterial community composition. Furthermore, sampled drinking water from the distribution system had a specific community composition related to either plant A or B, which could be used to determine the origin of drinking water sampled from mixed zones in the distribution system. Overall, we conclude that stored drinking water is highly susceptible to microbial deterioration. In addition, ATP and flow cytometry cell counts are poor indicators for microbial regrowth in the distribution system and citizen scientists seem able to reliably sample drinking water for microbial analyses.

## Introduction

In contrast to many other countries, drinking water in the Netherlands is distributed without a disinfectant residual [1], preventing odour/taste issues due to a disinfectant

**Data availability statement:** All relevant data are within the paper and its Supporting Information files.

**Funding:** The research was funded by the joint research program of the ten Dutch drinking water companies and one Flemish drinking water company.

**Competing interests:** The authors have declared that no competing interests exist.

residual. Microbial growth in Dutch drinking water distribution and premises plumbing systems is restricted by limiting the concentration of biodegradable organic carbon, which results in high quality drinking water [2]. Despite this high quality, microbial growth occurs during transport of drinking water to the consumer in the distribution and premises plumbing system [3–7], which can result in the growth of opportunistic pathogens that pose a threat to public health, aesthetic complaints (occurrence of invertebrates visible to the human eye and odor/taste) and technical complaints (pressure loss due to clogging of water meters with invertebrate biomass or corrosion) [8]. Some of these risks seem to be higher in the premises plumbing system due to longer retention times and a higher surface-to-volume ratio and water temperature [9]. It is common practice in the Netherlands that consumers refill their containers (i.e., bottles, water cookers, coffee reservoirs, etcetera) with drinking water from the tap. This offers both environmental and costs advantages as drinking water is much cheaper than bottled water and the number of water bottles used in the Netherlands is low compared to other countries [10–12]. However, the impact of storing drinking water in these bottles on the microbial water quality remains unknown.

Different studies investigated the influence of drinking water distribution on adenosine triphosphate (ATP), bacterial cell numbers and bacterial community in drinking water [3,6,7,13–18]. ATP is produced by all living organisms in their cells when consuming an energy source and is, therefore, an indicator for active biomass [19]. Most of these studies showed no influence of distribution on the ATP concentration, bacterial cell numbers or the bacterial community composition in drinking water, irrespective of the source water used (groundwater versus surface water) or disinfection strategy applied (chlorination, chloramination, or no disinfectant residual). In contrast, others demonstrated an increase in cell numbers in drinking water during distribution or a change in bacterial community composition [20, 21]. Fewer studies have investigated the influence of premises plumbing systems in private houses on ATP concentration, bacterial cell counts and bacterial community composition. When drinking water with a disinfectant residual was distributed, stagnation in premises plumbing system led to decreased concentration of the disinfectant residual, a 2-log increase in cell numbers, a decreased bacterial diversity and aberrant community composition [22,23]. In unchlorinated drinking water systems, it was observed that the ATP concentration and bacterial cell numbers only slightly increased (< 1-log increase) during overnight stagnation in the premises plumbing systems of different households [24,25]. Due to the lack of a statistical data evaluation in those two studies, it cannot be concluded whether overnight stagnation of unchlorinated drinking water in premises plumbing system significantly affected microbial biomass.

Drinking water quality in the Netherlands is routinely monitored by the drinking water companies for legal microbial parameters related to regrowth (heterotrophic plate counts at 22°C (HPC), *Aeromonas* plate counts and *Legionella* plate counts) and recurrently for additional non-legal parameters (e.g., ATP, AOC, biomass production potential, cell counts). The microbial drinking water quality in the Netherlands has been extensively studied in the past [8], and it was observed that general microbial drinking water quality parameters (e.g., ATP, cell counts, *Aeromonas*, HPC),

biological stability parameters (e.g., AOC, biofilm formation rate) and bacterial community composition differ between different production locations [6,7,17,26,27]. This implies that the effect of drinking water storage on microbial water quality might differ between drinking water treatment plants. None of the microbial data gathered from these monitoring programs, however, is actively communicated to the drinking water consumers. Not communicating such data to drinking water consumers might result in the observed "gap" or "distance" between water (companies) and citizens (customers) that has been reported to exist by a 2014 water governance assessment in the Netherlands [28]. The new EU Drinking Water Directive (2020/2184) emphasizes the importance of transparency regarding water quality information. Aiming to increase and maintain consumer confidence in water quality, it mandates that consumers must have access to up-to-date and comprehensible information about their drinking water quality. Over the past decade, a "new dawn" of citizen science can be witnessed, studies where the public actively participates in the generation of science-based knowledge. Such studies show the potential and limitation of involving citizen scientists (e.g., [29,30]). One of the issues involving citizen scientists in microbial drinking water quality is whether citizen scientists can provide reliable microbial sampling and analysis of drinking water and, thus, if they can be involved in monitoring the drinking water quality at the tap.

The main goal of our research was to determine how vulnerable drinking water is to deterioration. In addition, we also evaluated whether ATP, cell counts and community composition can be used as indicators for regrowth, and the feasibility of involving citizen scientists in reliable monitoring of the microbial drinking water quality. This was done by determining the influence of (i) drinking water transport in the distribution system, (ii) overnight stagnation in the premises plumbing system and (iii) storage in bottles on a range of microbial water quality parameters. The research was conducted in collaboration with citizen scientists in the city of Amsterdam and included households that retrieved drinking water from one of the two different treatment plants that supply the city with drinking water, as well as households that retrieved a mixture of drinking water from these two treatment plants. This approach made it possible to also determine the effect of mixing drinking water types on the microbial water quality.

## Materials and methods

### Sampling locations and citizen scientists

Drinking water for Amsterdam is produced at two different locations: treatment plant A and B. Both treatment plants treat surface water using precipitation/coagulation, rapid sand filtration, ozonation, softening, active carbon filtration and slow sand filtration. Dune infiltration after rapid sand filtration is an extra treatment step at plant A that is not incorporated at plant B. Plant A distributes drinking water to the Western part of Amsterdam, plant B to the Eastern part. The center part of the city receives drinking water from plant A and B.

In a previous paper we described how citizen scientists were recruited and subsequently selected for this study [31]. In total, 43 citizen scientists participated in the study and they lived across the inner city of Amsterdam (S1 Fig). Eighteen of them received drinking water solely from treatment plant A, 17 solely from treatment plant B and 8 received drinking water either from plant A or plant B depending on the time of the day. The drinking water samples were taken by the citizen scientists in June 2016. Before sampling the drinking water from their home, citizen scientists watched an instruction video on how to sample drinking water, inoculate and incubate the dip slides, and count the colonies. This video was shown during an initial meeting and remained accessible via a webpage dedicated to this project during the runtime of the project. All citizen scientists sampled drinking water at the kitchen tap in their household after overnight stagnation in the premises plumbing system and were, thus, taken in the morning before drinking water was used for other purposes, e.g., toilet flushing, showering, cleaning or drinking. They sampled drinking water directly after opening the tap (first flush) and after flushing for 5 min, the latter represents a drinking water sample from the distribution system. Next to these drinking water samples, the citizen scientists were invited to take a water sample of their own interest. Thirty of the 43 citizen scientists chose to sample drinking water that was stored in different containers (e.g., plastic bottles, glass bottles, coffee

machine reservoirs). The other 13 citizen scientists also sampled the drinking water after stagnation and flushing but their own interest sample was water from other sources than drinking water. These 13 non-drinking water samples were omitted from the data analyses presented in this publication. Furthermore, drinking water samples were also taken directly at both treatment plants, but by trained personnel in June 2016. The drinking water company Waternet approved access to their drinking water locations. A small portion of the drinking water samples were immediately used by the citizen scientists to determine bacterial and fungal plate counts, whereas the remainder of the drinking water samples were stored and transported to the lab at 4°C and processed within 24h. The ATP concentrations, cell numbers and 16S rRNA gene sequences were determined at the laboratory of KWR.

## Plate counts, ATP and cell numbers

The citizen scientists determined plate counts by inoculating an agar dip slide for bacteria and an agar dip slide for fungi (Sani-Check BF, Biosan Laboratories Inc, Warren, MI, USA) with drinking water that was sampled after 5 min of flushing and with the stored drinking water according to the manufacturer protocol. The dip slides were incubated for seven days at room temperature in the houses of citizen scientists, after which the colony forming units were counted by them.

The total ATP concentration, as a measure for active biomass, was determined in duplicate in all drinking water samples by measuring the amount of light produced in the luciferine-luciferase assay as previously described [6]. In a previous study, it was shown and explained that free ATP is negligible in drinking water samples from the Netherlands [6], meaning that total ATP concentrations determined in this study are a reliable measure for active biomass. The detection limit of the ATP-assay was 1.0 ng ATP/l. The total and membrane-intact cell counts were determined using a flow cytometer as described before [32].

Differences in ATP concentrations in flushed samples between the treatment plants were statistically tested. The Shapiro-Wilkinson test of normality demonstrated that the ATP concentrations were normally distributed ($p > 0.05$) and the Levene's test showed homogeneity of variance ($p > 0.05$). Subsequently, one-way analysis of variance (ANOVA) with Bonferroni post-hoc test was done. Furthermore, differences in ATP concentrations in flushed and direct drinking water samples and between direct and stored drinking water samples were also statistically analyzed. In this case the Shapiro-Wilkinson test of normality demonstrated that the ATP concentrations in these groups were not normally distributed ($p < 0.05$). Consequently, the non-parametric Mann-Whitney test was performed to compare these sample groups. In all cases, a p-value of 0.05 was used to accept ($p \geq 0.05$) or reject ($p < 0.05$) the null hypothesis.

## DNA isolation and community analysis

The drinking water samples (500 ml) were vacuum filtered through polycarbonate track-edge membrane filters with a diameter of 50 mm and a pore size of 0.2 µm (Sartorius; Goettingen, Germany). Filters were transferred to a bead tube of the MoBio PowerBiofilm Kit (MoBio; Carlsbad, USA) containing 350 µl of Solution BF1 from the kit. Samples were subsequently processed to isolate DNA and remove PCR-inhibitors according to the supplier's protocol and purified DNA was finally eluted in 100 µl elution buffer BF7.

Every drinking water sample was processed in duplicate to generate 16S rRNA gene amplicons using previously described 515F and 806R primers (containing Illumina adapter overhangs as described by Illumina) targeting the V4 hyper variable region of the 16S rRNA gene [33]. Triplicate amplicon reactions were pooled and 25 µl of this mixture was cleaned, indexed and sequenced as described in the Illumina MiSeq 16S Metagenomic sequencing library preparation protocol (https://support.illumina.com/content/dam/illumina-support/documents/documentation/chemistry_documentation/16s/16s-metagenomic-library-prep-guide-15044223-b.pdf, June 2016). The final amplicon concentration loaded on the MiSeq system was 4 pM supplemented with 10% PhiX (Illumina; San Diego USA) to add diversity. Negative controls were included in every experiment to monitor the presence of contaminating DNA. Version 1.37.0 of the MOTHUR software package (Schloss et al. 2009) was used to process all MiSeq datasets using the procedure previously described

(Kozich et al. 2013) and summarized in the MiSeq standard operating procedure (http://www.mothur.org/wiki/MiSeq_SOP, June 2016). The R1 sequences were quality filtered by removing sequences containing ambiguous bases and sequences with an average quality score below 35. Paired-end reads were first assembled into contigs. The quality of the quality filtered R1 reads, and the assembled reads were further improved by (i) removing sequences containing homopolymers of minimal 8 nt, (ii) removing sequences with ambiguous bases and (iii) by using a 1% precluster error. The remaining sequences were aligned to the Silva 16S RNA gene sequence database (Pruesse et al. 2007) and chimeric sequences were removed using UCHIME (Edgar et al. 2011). The number of 16S rRNA gene sequences per sample obtained after trimming varied between 10,156 and 98,652, with an average of 57,772 ± 16,953. Sequences were classified by comparison to the Ribosomal Database Project sequences (Cole et al. 2009) using a minimal confidence score of 80%. Sequences were first clustered into Operational Taxonomic Units (OTU) at a 3% dissimilarity level definition and taxonomically classified using trainset 9 of the RDP Hierarchy Browser.

The 16S rRNA gene read count data were first transformed into bacterial relative abundance. The relative abundance of taxa is calculated based on the proportional representation of each taxon within the total community. This is necessary because with the 16S rRNA gene sequencing method, the total number of reads differs between samples, making it impossible to reliably compare absolute abundances of the taxa. The relative abundance is derived from the number of 16S rRNA gene sequences (amplicon reads) assigned to each taxon, normalized by the total number of sequences in a sample. The online tool Microbiome Analyst, with default settings, was used for determining bar graphs at the taxonomic level (Dhariwal et al., 2017). The program PRIMER-e V7 (www.primer-e.com) was used to determine differences in bacterial community composition between samples by calculating Bray-Curtis dissimilarities, which were subsequently used as input for principle coordinates analysis (PCoA). A PERMDISP was done to test whether within-group dispersion is equivalent among the groups, followed by permutational multivariate analysis of variance (PERMANOVA) was done on the constrained axes used in ordination. A p-value of 0.05 was used as cut-off value for statistical significance.

## Results

### Sampling performance by the citizen scientists

In total 43 citizen scientists were selected in our study to participate in sampling and analyzing drinking water from the kitchen tap of their household. The ATP concentrations of the drinking water samples taken by the citizen scientists after 5 min flushing (representing drinking water from the distribution system) and at the treatment plant are shown in Fig 1. The ATP concentration in the treated water at plant A was 1.0 ng ATP/l and at plant B 4.3 ng ATP/l. The ATP-concentrations from citizen flushed samples from Plant A's distribution zone varied between < 1.0 and 2.2 ng ATP/l with an average of 1.3 ± 0.8 ng ATP/l (Table 1). The ATP-concentrations from citizen flushed samples from Plant B's distribution zone varied between < 1.0 till 3.5 ng ATP/l, with an average of 2.1 ± 1.0 ng ATP/l. These ATP concentrations are well in the range of what is normally measured in drinking water from the distribution system of both plants in Amsterdam (pers. comm. Marco Dignum). In addition, none of the drinking water samples taken from the tap by the citizen scientists had aberrant cell numbers, colony counts or bacterial community composition (data described below), compared to the samples taken by other citizen scientists or to the samples taken at the treatment plant by trained personnel.

### Influence of distribution, overnight stagnation and storage of drinking water

The ATP concentration remained relatively stable during distribution of the drinking water from treatment plant A to the households in the distribution system of this plant (Table 1, Fig 2). In addition, the average intact and total cell numbers were slightly lower in the distributed drinking water compared to the treated water of plant A. The ATP concentration in the treated water of plant B was twice as high as the average ATP concentration in the distributed drinking water from this plant. At all locations the distributed drinking water contained lower ATP concentrations than in the treated water of plant

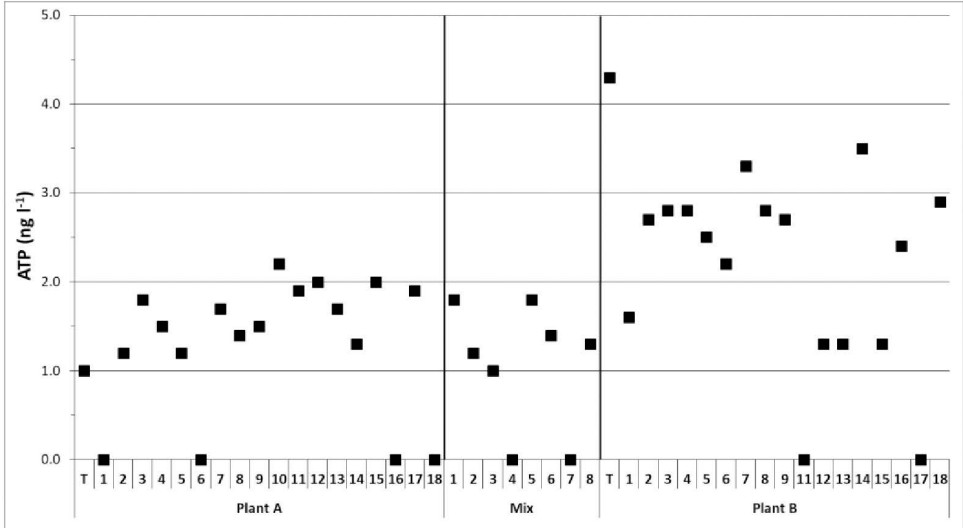

**Fig 1. The ATP concentrations of the drinking water sampled.** The drinking water samples at treatment plant A and B (T) were sampled by professional laboratory personnel and the drinking water samples at the tap were sampled by citizen scientists after 5 min flushing at the kitchen tap in their household (supplied with drinking water from plant A, plant B or from plant A and B, mix).

**Table 1. Average and standard deviation of ATP concentration, intact cell numbers, total cell numbers, ATP per intact cell and colony forming units of bacteria and fungi in drinking water from treatment plant A, treatment plant B and a mix of plant A and B.**

| | | ATP (ng l⁻¹) | | Intact cells (N ml⁻¹) | | Total cells (N ml⁻¹) | | ATP per cell (ng cell⁻¹)[a] | | Bacteria (cfu stick⁻¹) | | Fungi (cfu stick⁻¹) | |
|---|---|---|---|---|---|---|---|---|---|---|---|---|---|
| Type | Plant | Average | Stdev | Average | Stdev | Average | Stdev | Average | Stdev | Average | Stdev | Average | Stdev |
| Treated | A | 1.0 | – | $6.7 \times 10^4$ | – | $7.3 \times 10^4$ | – | $1.5 \times 10^{-8}$ | – | – | – | – | – |
| Flushed | A | 1.3 | 0.8 | $6.0 \times 10^4$ | $1.1 \times 10^4$ | $6.6 \times 10^4$ | $1.4 \times 10^4$ | $2.9 \times 10^{-8}$ | $8.0 \times 10^{-9}$ | 0.4 | 4.6 | 0.1 | 2.2 |
| Direct | A | 2.1 | 1.2 | $6.8 \times 10^4$ | $1.4 \times 10^4$ | $7.8 \times 10^4$ | $1.6 \times 10^4$ | $3.3 \times 10^{-8}$ | $1.6 \times 10^{-8}$ | – | – | – | – |
| Stored | A | 96.2 | 139.0 | $2.7 \times 10^5$ | $2.8 \times 10^5$ | $4.7 \times 10^5$ | $4.1 \times 10^5$ | $4.8 \times 10^{-7}$ | $9.7 \times 10^{-7}$ | 259.0 | 21.9 | 5.8 | 308.3 |
| Treated | B | 4.3 | – | $9.2 \times 10^4$ | – | $1.1 \times 10^5$ | – | $4.6 \times 10^{-8}$ | – | – | – | – | – |
| Flushed | B | 2.1 | 1.0 | $8.4 \times 10^4$ | $1.9 \times 10^4$ | $1.1 \times 10^5$ | $7.0 \times 10^4$ | $2.8 \times 10^{-8}$ | $8.6 \times 10^{-9}$ | 0.4 | 6.7 | 0.2 | 3.6 |
| Direct | B | 2.9 | 1.8 | $1.1 \times 10^5$ | $6.6 \times 10^4$ | $1.7 \times 10^5$ | $1.6 \times 10^5$ | $2.9 \times 10^{-8}$ | $1.3 \times 10^{-8}$ | – | – | – | – |
| Stored | B | 64.5 | 76.1 | $3.0 \times 10^5$ | $3.5 \times 10^5$ | $4.1 \times 10^5$ | $5.1 \times 10^5$ | $2.8 \times 10^{-7}$ | $2.1 \times 10^{-7}$ | 64.2 | 36.2 | 0.3 | 5.4 |
| Flushed | M | 1.1 | 0.7 | $5.6 \times 10^4$ | $1.0 \times 10^4$ | $6.2 \times 10^4$ | $1.2 \times 10^4$ | $2.7 \times 10^{-8}$ | $8.2 \times 10^{-9}$ | 0.9 | 5.2 | 0.2 | 3.6 |
| Direct | M | 1.7 | 0.9 | $6.0 \times 10^4$ | $1.7 \times 10^4$ | $6.9 \times 10^4$ | $2.1 \times 10^4$ | $3.4 \times 10^{-8}$ | $1.9 \times 10^{-8}$ | – | – | – | – |
| Stored | M | 20.1 | 31.6 | $9.5 \times 10^4$ | $5.9 \times 10^4$ | $1.2 \times 10^5$ | $7.0 \times 10^4$ | $1.9 \times 10^{-7}$ | $1.9 \times 10^{-7}$ | 4.0 | 8.9 | 0.2 | 5.2 |

B (Table 1, Fig 2), demonstrating that active biomass decreased during distribution. Furthermore, the intact and total cell numbers were comparable between treated and distributed drinking water from plant B (Table 1).

The average ATP concentration, membrane-intact and total cell numbers were higher in drinking water sampled directly from the tap after overnight stagnation than in drinking water sampled at the tap after 5 min flushing (Table 1), but differences were not statistically significant (Mann-Whitney, p > 0.05). The influence of overnight stagnation on ATP or bacterial cell numbers differed considerably per location. At 25 of the 43 locations the ATP concentration was higher in drinking water directly sampled from the tap after overnight stagnation in the premises plumbing system than in drinking water sampled from the tap after 5 min flushing (Fig 2). At 19 of these 25 locations the ATP increased with more than 20%,

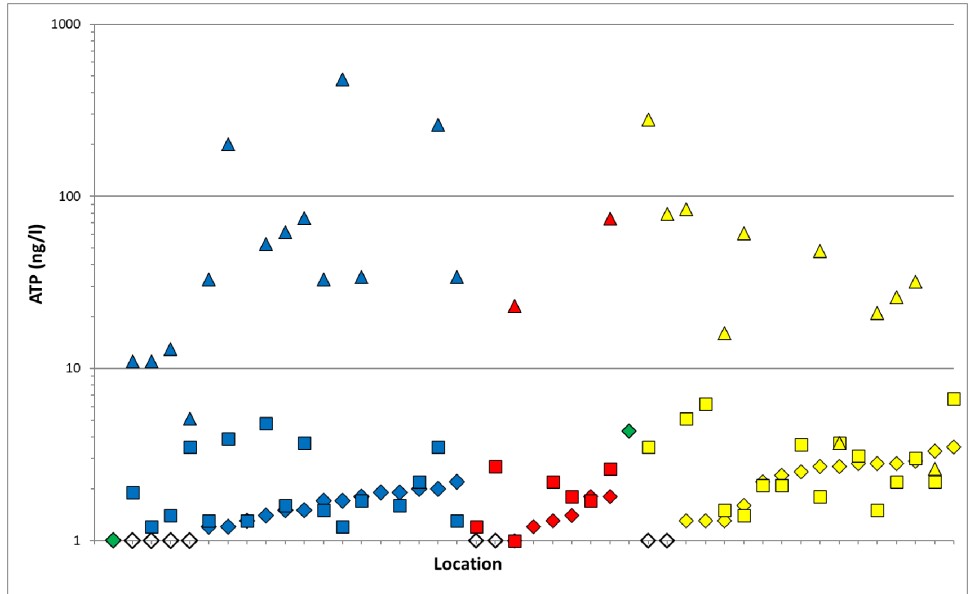

**Fig 2. ATP-concentration in drinking water sampled at two plants and 43 locations in the distribution systems.** Drinking water came from Plant A (blue symbols), Plant B (yellow symbols), mixed zone of Plant A and B (red symbols) or was treated water sampled at plant A or B (green symbols). Symbols: ◆, Drinking water sampled at the kitchen tap after 5 min flushing (representing drinking water from the distribution system); ■, drinking water sampled directly after opening the kitchen tap after overnight stagnation; ▲, drinking water stored in bottles. Open symbols: ATP concentrations lower than detection limit of 1 ng/l ATP.

indicating microbial growth in the premises plumbing system during overnight stagnation at these 19 locations. Furthermore, at 15 locations the ATP concentrations were lower in the directly tapped drinking water samples than in the flushed drinking water samples with eight locations showing a more than 20% decrease in ATP concentration, indicating active biomass decay during stagnation at those locations. The remaining three locations showed the same ATP concentration in directly tapped and flushed drinking water, suggesting balance between biomass growth and decay.

The number of total and membrane-intact cells and total cells were higher in directly tapped drinking water samples after overnight stagnation than in the 5 min flushed drinking water samples in 37 (total cells) or 35 (membrane-intact cells) of the 43 samples, but only at around ten locations the membrane intact or total cell numbers increased with more than 20% (S2 and S3 Fig). At four (total cells) to six (membrane-intact cells) locations, intact and total cell numbers were lower in directly tapped drinking water samples than in flushed samples, but only at one location was this difference more than 20%. The remaining two locations showed the same membrane-intact and total cell numbers in directly tapped and flushed drinking water samples.

The average ATP concentration and cell numbers were significantly higher in the drinking water samples that have been stored in containers than in drinking water sampled after 5 min flushing at the tap (Mann-Whitney, $p < 0.05$) (Table 1, Fig 2, S2 and S3 Figs). On average the ATP increased by a factor 12–74, whereas the total or intact cell numbers only increased by 1.6 to 7.2. As a result, the ATP per cell ratio was much higher in stored drinking water than in flushed drinking water. Most locations showed more than 20% higher ATP concentrations or cell numbers in stored drinking water than in flushed drinking water samples from the tap, demonstrating microbial growth occurred during drinking water storage in different containers. The results from the bacterial and fungal plate counts, that were determined by the citizen scientists, demonstrated that stored drinking water also had higher average bacterial and fungal plate counts than drinking water sampled after 5 min flushing of the tap (Table 1). Particularly the increase in bacterial plate counts in stored drinking water

compared to flushed drinking water from the tap was much larger (up to 724 times) than observed for ATP and especially cell numbers counted with flow cytometer. This result is an indication that not only the biomass increased during drinking water storage but that the community also shifted towards one harboring more bacteria capable of growing on the agar dip slides for bacteria used in our study.

The bacterial community composition was elucidated using microbial profiling of the bacterial 16S rRNA gene sequences obtained with next generation sequencing. The number of sequences analyzed, OTUs observed, and the Shannon diversity indices are shown in S1 Table (for each sample) and Table 2 (average values per water type and plant). In general, 25,000–95,000 16S rRNA gene sequences were analyzed per sample and the average sequences obtained did not differ noticeably per water type and treatment plant. The observed OTU numbers and the Shannon diversity index were highest in drinking water sampled after 5 min flushing of the tap and were slightly lower in drinking water samples that were taken directly from the tap after overnight stagnation (Table 2). A substantial decrease in OTU numbers and Shannon index was observed in the drinking water samples that were stored in containers (Table 2).

The relative abundance of the different dominant bacterial classes did not change considerably during transport in the distribution system from the treatment plant to the households of the citizen scientists (Fig 3A). Bacteria belonging to unclassified taxonomic classes prevailed in these samples, indicating that a large part of the bacterial community remained unknown even at class level. Besides these unclassified classes, *Proteobacteria* and *Acidobacteria* GP6 also dominated the treated and distributed drinking water samples. Overnight stagnation in the premises plumbing system did not have a large impact on the bacterial community at class level either (Fig 3B), although the relative abundance of bacteria belonging to unclassified classes and *Acidobacteria* GP6 slightly decreased compared to the flushed drinking water samples. This decrease was accompanied by an increase in the relative abundance of *Proteobacteria* classes and *Actinobacteria*. A substantial change in the bacterial community at class level was observed in the drinking water samples that were stored in containers (Fig 3C). Compared to the flushed or directly sampled drinking water from the tap, the relative abundance of bacteria belonging to unclassified classes in these stored drinking water samples were very low, whereas *Proteobacteria* strongly dominated most of the stored drinking water samples. Furthermore, the relative abundance of *Sphingobacteria* and *Actinobacteria* also increased in several stored drinking water samples.

The bacterial drinking water communities were also compared at OTU level using principal coordinate analysis of the calculated Bray Curtis weight distance matrix. A comparison of all analyzed drinking water samples (treated, flushed, directly tapped and stored drinking water) showed that the bacterial community in treated, flushed and directly tapped

**Table 2. Average and standard deviation of number of sequences and OTUs, and Shannon index for the different drinking water types from treatment plant A, treatment plant B and a mix of plant A and B.**

| Type | Plant | # Sequences | | # OTUs | | Shannon index | |
|---|---|---|---|---|---|---|---|
| | | Average | Stdev | Average | Stdev | Average | Stdev |
| Treated | A | 64058 | | 3640 | | 6.9 | |
| Flushed | A | 54406 | 10239 | 3641 | 421 | 7.0 | 0.1 |
| Direct | A | 55353 | 13706 | 3055 | 474 | 6.5 | 0.3 |
| Stored | A | 64390 | 19207 | 822 | 451 | 3.1 | 0.8 |
| Treated | B | 25060 | | 2394 | | 6.8 | |
| Flushed | B | 53547 | 20532 | 3567 | 869 | 6.9 | 0.3 |
| Direct | B | 58871 | 16111 | 3032 | 934 | 6.2 | 1.2 |
| Stored | B | 67215 | 21229 | 2285 | 1383 | 4.8 | 1.8 |
| Flushed | M | 50905 | 18683 | 3440 | 675 | 7.1 | 0.1 |
| Direct | M | 50863 | 11898 | 2862 | 262 | 6.5 | 0.3 |
| Stored | M | 58878 | 30760 | 1247 | 1042 | 4.1 | 1.2 |

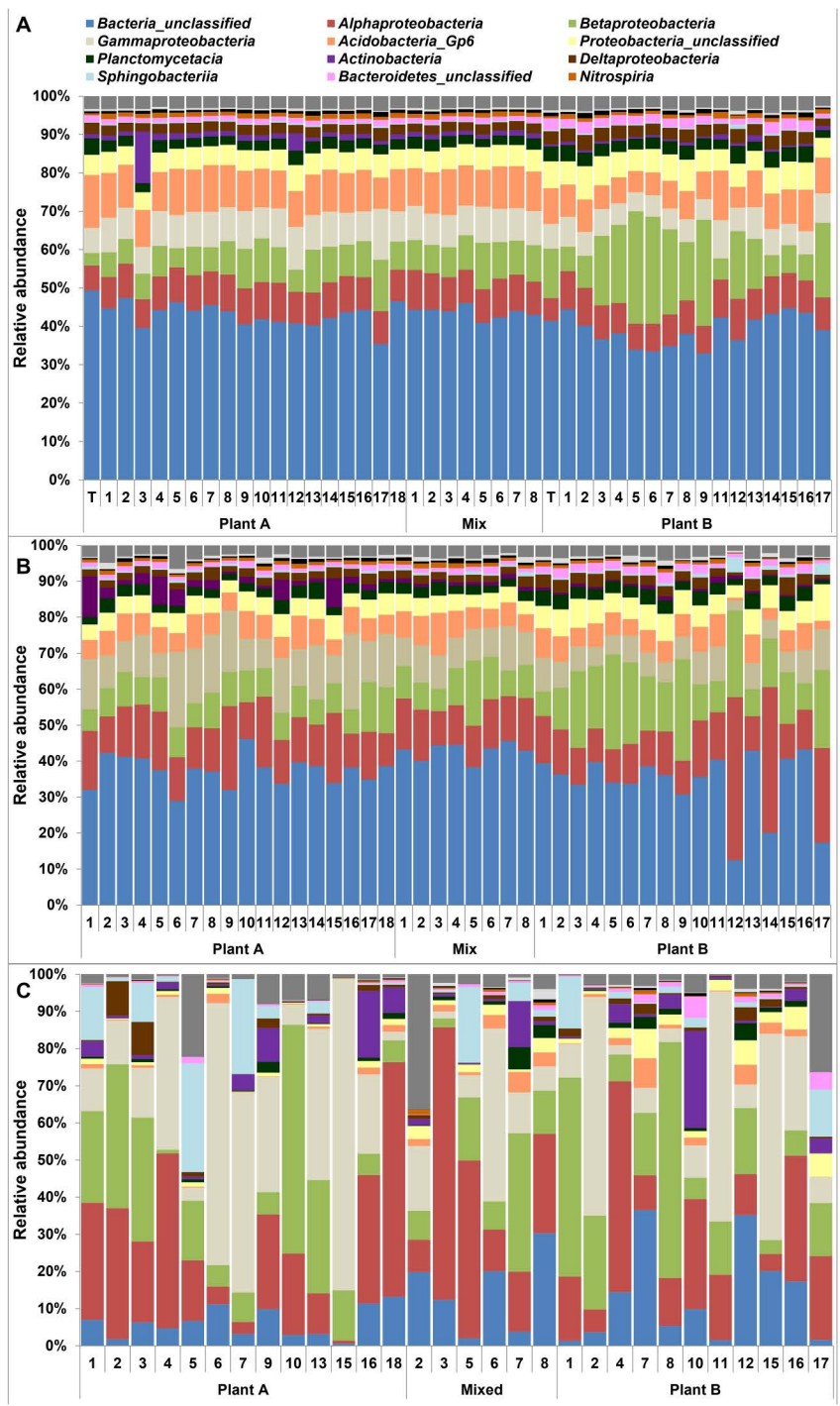

**Fig 3. Taxaplot of the prokaryotic community members at class level observed in the drinking water samples.** The drinking water samples were taken from the tap after 5 min flushing **(A)**, directly after opening the tap after overnight stagnation **(B)** and after drinking water was stored in a container **(C)**. Drinking water samples were retrieved from the distribution area of Plant A, Plant B and the mixed zone of Plant A and plant B.

drinking water clustered more together compared to the stored drinking water samples ([Fig 4A](https://doi.org/10.1371/journal.pone.0335138.g004)). PERMANOVA showed that the bacterial community in the stored drinking water samples were significantly different from those in the drinking water sampled at the plant or the tap. However, the PERMDISP results demonstrated that the homogeneity of dispersion between stored drinking water samples and drinking water sampled at the plant or the tap were different ($p < 0.05$). Moreover, the stored drinking water samples scattered over the whole PCoA plot, irrespective of the source of the drinking water (treatment plant A, B or a mix of plant A and B). Additional analysis on the average centroid distances confirmed that the average centroid distances of the stored drinking samples (54.3%) was significantly higher than those of the direct sampled drinking water (34.2%) and flushed drinking water samples (30.8%) ($p < 0.05$). These results together implied that

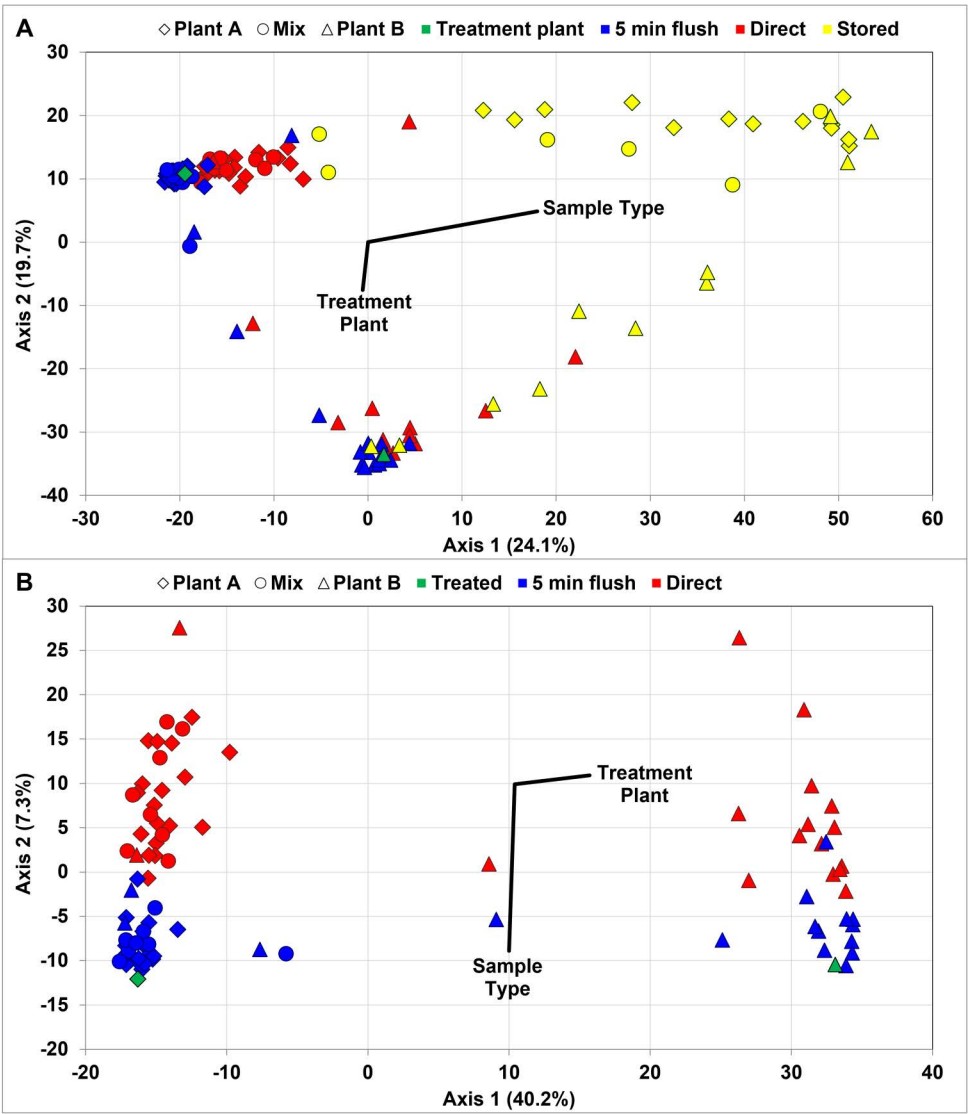

**Fig 4. Principal Coordinates Analysis (PCoA) plot of the OTU composition in the drinking water samples.** Drinking water samples from the treatment plant (green) and drinking water taken directly (red) or after 5 min flushing (blue) at the tap and after storage in containers (yellow). The drinking water came from treatment plant A, treatment plant B or were taken at a mixed zone in the distribution system where drinking water came from plant A and B. Panel **A**: all samples; Panel **B**: stored drinking water samples are omitted.

extensive changes occurred in the bacterial OTU community composition when drinking water was stored in containers, but also that these changes were rather specific for each household that stored the drinking water.

To better observe changes in the bacterial OTU community during drinking water transport in the distribution system and overnight stagnation in the premises plumbing system, a PCoA plot was also made without the stored drinking water samples (Fig 4B). These results showed that the bacterial community also differed between drinking water taken directly from the tap or after 5 minutes flushing. PERMDISP results showed homogeneity of dispersion between the direct and 5 min flushed drinking water samples (p > 0.05) and PERMANOVA confirmed that the observed changes were statistically significant (p < 0.05). Furthermore, we observed that the dominant bacterial OTU community composition of the treated water at plant A is comparable to the dominant bacterial OTU community composition in all drinking water samples from the distribution system (i.e., flushed samples) where drinking water from plant A is distributed. Similar results were observed for the flushed drinking water samples from the distribution system and the treated water of plant B. These results, thus, demonstrated that the dominant bacterial OTU community composition remained stable during transport in the distribution system of plant A and B. Furthermore, the drinking water samples originating from plant A clustered in general separate from the drinking water samples originating from plant B. In contrast, the bacterial community in drinking water sampled from households where drinking water either came from plant A or plant B clustered with the drinking water samples originating from plan A. PERMDISP showed that the homogeneity of dispersion was equal between drinking water samples from plant A, plant B and locations that received drinking water from plant A or B (p > 0.05). Pairwise PERMANOVA showed that the community composition of drinking water originating from plant B was significantly different from those in drinking water originating from plant A or in drinking water sampled from households that either received drinking water from plant A or plant B (p < 0.05).

The PCoA results also showed that the flushed and directly tapped drinking water samples clustered based on the origin of the water (treatment plant A or B) at the first coordinate and on water type (flushed or directly tapped) at the second coordinate (Fig 4B). This indicates that the origin of the drinking water was a stronger driver for changes in the dominant bacterial OTU community composition than overnight stagnation in the premises plumbing system. Still, the results clearly showed that changes in the dominant bacterial OTU community composition were larger during overnight stagnation in the premises plumbing system of a household, than during transport in the distribution system from the treatment plant to the water meter (i.e., inlet) of the household.

The overall results from the community analysis, thus, demonstrated a substantial change in the community composition, OTU numbers and the Shannon diversity during storage and use of drinking water in containers, which was also accompanied in a change from unclassified to classified bacterial classes. We, therefore, also examined if drinking water storage resulted in an increase in highly dominant (≥ 10% relative abundance) OTUs in the bacterial community and, if so, to what genera these OTUs belong (S2 Table). An OTU with a relative abundance above 10% in the drinking water sample taken after 5 min flushing or directly at the tap was observed at respectively three or ten of the 30 households. In contrast, 58 OTUs with a relative abundance above 10% were observed in stored drinking water at 26 of the 30 households, with most of these samples (19 out of these 26 households) showing more than one OTU having a relative abundance above 10%. Because the same OTU sometimes dominated in containers from multiple households, only 36 of these 58 OTUs were unique

Twenty-one of the 58 OTUs observed in stored drinking water with a relative abundance above 10% were not observed in the flushed or directly tapped drinking water at the same location (Table S2). This indicates that these bacteria were absent or only present at very low numbers in the drinking water used to fill the containers before storage, and that they grew to dominant levels during storage. In addition, 19 other of these 58 OTUs in stored drinking water already slightly increased in relative abundance during overnight stagnation in the premises plumbing system (to a maximum 0.34% relative abundance), but in general this increase was negligible compared to the large increase during drinking water storage (from 0.0026–0.34% relative abundance in directly tapped drinking water to 10.5–58.8% relative abundance in

stored drinking water) (S2 Table). Still, it might demonstrate that growth of these bacteria already occurred during overnight stagnation and continued when the drinking water was stored for a longer period in a container. Three OTUs that had a relative abundance above 10% in stored drinking water were observed in four different stored water samples, two OTUs in three different stored waters samples and nine in two different stored water samples. The rest (22 OTUs) were only observed in one of the stored drinking water samples. This result demonstrates that in each stored drinking water in general (a) different OTU(s) dominated.

These most dominant OTUs observed in stored drinking water could for a large part be assigned to the genus level (21 out of 36) (S2 Table). The remaining OTUs could be assigned to the family (8 OTUs), order (4 OTUs) or class (3 OTUs) level. The dominant OTUs that could be assigned to the genus level belonged to 15 different genera with *Acinetobacter* (5 OTUs) and *Pseudomonas* (3 OTUs) being most often observed. At six households, drinking water storage in a container resulted in a single OTU having a relative abundance above 50%. At two of these six households the same OTU prevailed in stored drinking water and this OTU belonged to the genus *Enhydrobacter*. In both households the relative abundance of this OTU already increased to 13.4% or 1.5% in drinking water during overnight stagnation, indicating that growth of this OTU started during drinking water stagnation. At the other four households the prevailing OTU in stored drinking water belonged either to the genus *Acidovorax*, *Acinetobacter*, *Methylobacterium* or *Pseudomonas*. These OTUs had not yet increased significantly in the drinking water during overnight stagnation and, thus, started to become dominant during drinking water storage.

### Influence of treatment plant

The ATP concentration and intact and total cell numbers in the treated water of plant A was lower than that of plant B (Table 1), indicating that the treated water of Plant B had a higher concentration of active biomass than that of Plant A. Moreover, the average ATP concentration and intact and total cell numbers in drinking water taken after 5 min of flushing were significantly higher in drinking water samples taken at households in the distribution system of plant B than taken at households in the distribution system of plant A ($p < 0.05$; one-way Anova with Bonferroni posthoc). Furthermore, the drinking water samples taken in the mixing zone of the distribution system, where drinking water from plant A and B were distributed, had ATP concentrations and cell numbers that were significantly different from samples taken in the distribution system of plant B ($p < 0.05$), but not from plant A ($p > 0.05$). This result implies that the mixing zone of plant A and B were mostly fed with drinking water from treatment plant A. In addition, it was observed that the colony forming units of bacteria and fungi in flushed drinking water were in general below the detection limit (1 cfu per dipslide) and, as a result, differences between the water types (i.e., drinking water from plant A, plant B or mixing zone) could not reliably be determined for these plate counts.

The relative abundance of 16S rRNA gene sequences belonging to the 14 most dominant classes showed that the bacterial community did not differ considerably at class level between drinking water sampled from the distribution system of plant A and B after 5 min flushing of the tap (Fig 3A). Only the relative abundance of *Acidobacteria* GP6 was higher in flushed drinking water from plant A compared to plant B, whereas the abundance of the *β-Proteobacteria* was lower. Most of the flushed drinking water samples from the mixing zone of the distribution system also seemed to have a lower relative abundance of *β-Proteobacteria*. The PCoA and PERMANOVA of the distance matrix (based on Bray Curtis) at OTU level showed that the bacterial OTU community composition of drinking water sampled from the distribution area of plant A was significantly different from plant B ($p < 0.05$) (Fig 4B). However, the bacterial OTU community composition of distributed drinking water sampled from the mixing zone of the distribution system was remarkably similar to the bacterial OTU community composition of plant A and no statistically significant differences between the bacterial communities could be observed ($p > 0.05$). This result indicates again that drinking water from treatment plant A, but not from treatment plant B, was distributed in the mixed zone of the distribution system when the distributed drinking water samples were taken in the morning at the kitchen tap.

## Discussion

### Citizen scientists

Our study involved citizen scientists for sampling the different drinking water types and determining the plate counts of bacteria and fungi from the sampled drinking water. A potential limitation of including citizen scientists for water quality monitoring is that citizen scientists might include errors because they are inexperienced and not well trained to sample water and do microbiological analyses. The collaboration with citizen scientists in water quality research is not new, with numerous examples of citizen involvement, particularly in determining surface water quality (e.g., [34,35]). However, publications on the involvement of citizen scientists in monitoring drinking water quality are scarce [36–41]. In one of those studies, basic chemical parameters such as pH, conductivity, chloride, nitrate and hardness were measured by trained students that sampled the drinking water in their homes [41]. The student results were within the limits of uncertainties, but for some parameters significant differences were observed with the laboratory results. In a study where citizen scientists monitored microbial water quality, they were recruited to sample biofilms from the showerhead in their homes and to analyze basic chemical parameters [36]. These authors did not elaborate on whether and how involved citizen scientists were trained and on the sampling and chemical data reliability of the citizen scientists. Low standard deviations for the chemical parameter means were observed, but high variability between samples for the microbial community analysis occurred. As a result, it remains difficult to conclude whether involvement of citizen scientists resulted in the collection of reliable microbial data in that study.

We compared the results from samples taken at each household by citizen scientists with the results from samples taken at the treatment plant by a professional sampling technician, as well as between citizen scientists. These comparisons indicate that no aberrant results for ATP, cell numbers, heterotrophic plate counts, and community composition were obtained for the drinking water samples taken after five minutes flushing by each of the 43 citizen scientists. Furthermore, the ATP concentrations and cell numbers are in the range that the drinking water company measures with their water quality monitoring program in these distribution systems. We, therefore, conclude that, when properly instructed, citizen scientists can reliably sample drinking water for ATP, cell count, community composition and heterotrophic plate count analysis. In the same project, we also analyzed the social impact of involving citizen scientists in drinking water research, but those results were published in a separate paper [31]. One of the results from those analyses was that a majority of the citizen scientists reported an increase in their confidence in both the drinking water quality and the drinking water company as a result of their participation in this project [31]. Based on these findings, we encourage drinking water companies to consider collaborating with voluntary citizen scientists for drinking water sampling and monitoring, since this opens possibilities to improve monitoring strategies for drinking water quality while simultaneously increasing consumer confidence in the safety and reliability of their drinking water.

### Drinking water residence in distribution and premises plumbing systems

We investigated how the microbial quality of drinking water changed during distribution, overnight stagnation in a premises plumbing system and storage in a container. The results demonstrated that the active biomass, cell numbers and community composition remained relatively stable during transport in the drinking water distribution system. Others have also reported that the active biomass concentration, bacterial cell numbers or the bacterial community composition in drinking water were hardly affected during transport in the distribution system, irrespective of source water (groundwater versus surface water) and disinfection strategy (chlorination, chloramination or no disinfectant residual) used [3,6,7,13–18]. However, several studies have demonstrated biofilm formation and growth of specific microorganisms (e.g., *Aeromonas*, coliforms, mycobacteria and *Legionella*) in drinking water during transport in the distribution system [4,7,32,42–46]. Studies have also shown that some opportunistic pathogens and heterotrophic and *Aeromonas* plate counts increased in drinking water during transport in the distribution system of Amsterdam [32,47]. Consequently, there seems to be an apparent

 

difference between the dynamics in the generic microbiological parameters and the dynamics of specific nuisance micro-organisms during distribution of drinking water in Amsterdam. This discrepancy between dynamics in general and specific microbiological parameters have also been reported for drinking water in the distribution systems of 28 different treatment plants in the Netherlands [7]. It was concluded from that study that *Aeromonas* and heterotrophic plate counts are more reliable indicators for regrowth in the distribution system than ATP or cell counts. As a result, the reliability of monitoring generic microbiological parameters, like total, membrane intact and HNA/LNA cell counts, and ATP as indicators for regrowth of nuisance organisms in drinking water distribution systems can be questioned. Consequently, shifting routine microbial drinking water quality monitoring from HPC, *Aeromonas* and *Legionella* plate counts to ATP or cell counts using flow cytometry, as suggested previously [48,49], is not recommended. It still also presses the need to develop other rapid indicators for regrowth problems in drinking water distribution systems, since plate counts methods are time consuming.

Storage of drinking water had the largest influence on the microbial quality, as evidenced by the substantial changes in biomass, cell numbers and community composition when drinking water was stored in containers. An onset of these changes could already be observed during overnight stagnation, which led to a modest increase in biomass and cell numbers and slightly affected the bacterial community composition at several households analyzed in our study. Many studies have shown that stagnation of drinking water resulted in altered microbial community and increased the risk of opportunistic pathogens growth like *L. pneumophila* or certain nontuberculous mycobacterial species (e.g., [9, 23, 50–55]). Most of these studies, however, were conducted on drinking water with a disinfectant residual. In those cases, stagnation leads to loss of disinfectant residual and, as a result, microbial growth is no longer inhibited. Since drinking water in the Netherlands is distributed without a disinfectant residual, the results from those studies cannot be compared with our study. Some studies have also investigated the effect of (overnight) stagnation on the microbial quality of drinking water without a disinfectant residual [24,54]. Those studies showed that the biomass and cell numbers in drinking water increased, and that the microbial community composition slightly changed during residence in the premises plumbing system. The latter observation was also made in our study, but in contrast to those two studies we observed that it depended on the household whether biomass and cell numbers increased or decreased in the drinking water during overnight stagnation in the premises plumbing system. This apparent discrepancy might have been caused by the low number of households investigated in the previous studies (n = 1 or n = 10) [24,54] and households where drinking water biomass and cell numbers decreased during overnight stagnation might have been missed. Another possible explanation for this discrepancy can be differences in plumbing materials, usage patterns or temperature in the households that were investigated in both studies.

The decrease in biomass and cell numbers we observed at several households might have been caused by the low BDOC concentrations in the drinking water in the Netherlands [26], resulting in microbial starvation during overnight stagnation. However, if this mechanism had been the cause behind the biomass decrease during overnight stagnation, biomass should have decreased at all households, since they all receive drinking water with low BDOC concentrations. Another possibility is that the households where a decrease in biomass and cell numbers were observed after overnight stagnation have a copper-based premises plumbing system. It is known that copper ions can act as a disinfectant and hence copper ions released in the biofilm and water might have elucidated microorganisms in the biofilm and drinking water (reviewed by [56]). It is still important to stress that at several households both active biomass and cell numbers increased in the drinking water during overnight stagnation, demonstrating that overnight stagnation can result in microbial growth in the drinking water premises plumbing system.

## Microbial drinking water quality deterioration during container storage

Due to the high drinking water quality and the distribution without a disinfectant residual, it is common practice in the Netherlands to consume water directly from the tap instead of bottled water. Many people in the Netherlands do this by refilling a (plastic) container with drinking water and, therefore, it was not surprising that for their own chosen water sample, most citizen scientists sampled a container they use to refill with drinking water. The results for the drinking water from these

containers strikingly showed that there was a strong increase in biomass and cell numbers concomitant with a decrease in diversity of the microbial community composition. It can be concluded that container storage had the largest influence on the microbial water quality compared to drinking water transport in the distribution system or overnight stagnation in the premises plumbing system. A likely cause for the large impact on the microbial drinking water quality during container storage is that people directly drink from these containers, resulting in transfer of saliva from the mouth into the container. Such saliva transfer results in introduction of bacteria from the oral microbiome and of relatively high BDOC concentrations in the water, thereby promoting growth of bacterial species with a high maximum specific growth rate. Our results showed that species belonging to the genus *Acidovorax*, *Acinetobacter*, *Pseudomonas*, *Methylobacterium* and *Enhydrobacter* (S2 Table) could strongly dominate (> 50% of the community) the drinking water in these containers. It has been demonstrated that *Pseudomonas* species in drinking water have a relatively high maximum specific growth rate, but low affinity compared to other drinking water microorganisms [57], which further support this hypothesis. Although storage of drinking water in these containers altered the microbial water quality, it remains questionable whether these alterations would also have a health impact. The results demonstrated that most of the genera that became dominant in these containers do not contain virulent pathogenic species, suggesting that the public health risk of the drinking water deterioration during storage in these containers was low.

To our knowledge, no previous studies have been published on the influence of drinking water storage in (bottle) containers on the microbial community. However, several studies have investigated dynamics in the microbial community of bottled mineral water during storage ([51–55]). The results from those studies show that storage of bottled water generally also resulted in increasing cell numbers and colony counts as well as a substantial change in bacterial community composition. In addition, Sala-Comorera et al. [54] observed that during storage of different brands of bottled waters the percentage of cultivable bacteria increased with a sharp decrease in diversity, which is in concordance with our observation for unchlorinated drinking water stored in a container. It is not surprising that results from bottled mineral water storage were comparable to our findings, since mineral water, like drinking water in the Netherlands, have low BDOC concentrations and do not contain a disinfectant residual. In contrast to our study, however, Sala-Comorera et al. [52] observed that most 16S rRNA gene sequences could be identified to the genus level directly after bottling, but that during storage unclassified genera became dominant.

## Conclusions

Overall, we conclude from our results that drinking water is a product prone to microbial quality deterioration during storage, comparable to other food products (e.g., vegetables, dairy products, meat) [58,59]. Quality deterioration during other food product storage can result in human health risks (due to growth of pathogens) or taste and odor issues [58,59]. Whether microbial quality deterioration during drinking water storage in containers can result in human health risks or taste and odor issues has not yet been investigated to the best of our knowledge, nor can they be inferred from our study. A more targeted approach, using specific detection methods for nuisance organisms, is needed before more definite conclusions can be drawn about possible public health or aesthetic impacts of drinking water storage in containers. Furthermore, generic microbial parameters like ATP concentration or cell counts determined by flow cytometry seem to be less suitable indicators for regrowth in the distribution system, whereas the microbial community composition can reliably be used to track the drinking water origin to a treatment plant. Finally, we conclude that citizen scientists seem to be able to reliably sample drinking water for microbial analyses, and we recommend their involvement when sampling drinking water from households.

## Supporting information

**S1 Fig. Map of Amsterdam with the locations of the citizen scientists participating in the study marked with blue circles.**
(DOCX)

**S2 Fig. Total cell numbers in drinking water sampled at 43 locations in the distribution system of treatment Plant A (blue symbols), Plant B (yellow symbols), mixed zone of Plant A and B (red symbols) and in treated water sampled at plant A or B (green symbols).**
(DOCX)

**S3 Fig. Membrane-intact cell numbers in drinking water sampled at 43 locations in the distribution system of treatment Plant A (blue symbols), Plant B (yellow symbols), mixed zone of Plant A and B (red symbols) and in treated water sampled at plant A or B (green symbols).**
(DOCX)

**S1 Table. Number of sequences, OTUs and the Shannon diversity index for each sample.**
(DOCX)

**S2 Table. Taxonomic classification of OTUs with ≥ 10% dominance in the community of flushed, direct or stored drinking water samples as well as their relative abundance in the other water types at the same location.**
(DOCX)

## Acknowledgments

The authors would like to thank the citizen scientists that participated in this project, Maarten Claassen and Leon Kors from Waternet for their help and discussions, and the staff of the microbiological laboratory of KWR for analyzing the samples. A major part of the outcomes presented in this paper was based on research financed by the Joint Research Program of the drinking water companies in The Netherlands.

## Author contributions

**Conceptualization:** Paul W.J.J. van der Wielen, Stijn Brouwer, Marco Dignum, Merijn Schriks.

**Data curation:** Paul W.J.J. van der Wielen, Stijn Brouwer, Merijn Schriks.

**Formal analysis:** Paul W.J.J. van der Wielen.

**Investigation:** Paul W.J.J. van der Wielen.

**Methodology:** Paul W.J.J. van der Wielen, Stijn Brouwer, Merijn Schriks.

**Supervision:** Stijn Brouwer, Marco Dignum, Merijn Schriks.

**Writing – original draft:** Paul W.J.J. van der Wielen.

**Writing – review & editing:** Stijn Brouwer, Marco Dignum, Merijn Schriks.

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
