## [Decision Letter · Decision Letter 0]

11 Jun 2025

Dear Dr. van der Wielen,

Thank you for submitting your manuscript to PLOS ONE. After careful consideration, we feel that it has merit but does not fully meet PLOS ONE’s publication criteria as it currently stands. Therefore, we invite you to submit a revised version of the manuscript that addresses the points raised during the review process.

Both reviewers found the work to be important and timely. However, there should be more emphasis on a standardization of the methodology and protocols used in the water testing by citizens. The conclusions should also be reconsidered and made more tentative, given the exploratory nature of the work. Pay particularly close attention to the comments of reviewer #1. There are issues with the consistency of the writing and the flow. The reviewers have included extensive notes and comments for recommendations to improvement of the text. These should be carefully considered and the changes incorporated into any resubmission.  

We look forward to receiving your revised manuscript.

Kind regards,

Theodore Raymond Muth

Academic Editor

PLOS ONE

Additional Editor Comments (if provided):

Reviewers' comments:

Reviewer's Responses to Questions

**Comments to the Author**

1. Is the manuscript technically sound, and do the data support the conclusions?

Reviewer #1: Yes

Reviewer #2: Partly

2. Has the statistical analysis been performed appropriately and rigorously?

Reviewer #1: Yes

Reviewer #2: Yes

3. Have the authors made all data underlying the findings in their manuscript fully available?

Reviewer #1: Yes

Reviewer #2: Yes

4. Is the manuscript presented in an intelligible fashion and written in standard English?

Reviewer #1: Yes

Reviewer #2: Yes

Reviewer #1: The paper addresses an important and timely topic: the vulnerability of non-chlorinated drinking water to microbial deterioration during distribution, stagnation, and storage, using a citizen science approach. This is highly relevant for public health monitoring and community engagement, particularly in regions like the Netherlands where drinking water is distributed without residual disinfectants. However, the manuscript, while methodologically sound in many respects, suffers from a few limitations in clarity and structure. These include inconsistencies in phrasing, unclear sample classification, and a lack of discussion around potential variability introduced by citizen-led procedures. The limitations of dipslide-based microbial counts, the need for standardization in sampling and incubation, and clarification of the analytical pipeline, should be addressed to strengthen the scientific rigor and readability of the manuscript. The conclusions should be more cautious given the exploratory nature and dataset limitations.

Abstract

Lines 15–17: The opening sentence is overly long and could be made more digestible.

Lines 20–21: The phrase “prokaryotic cell counts using flow cytometry, ATP concentrations and the prokaryotic community composition was determined” contains a subject-verb agreement error and could be clearer.

Line 23: Replace "demonstrated" with "showed" or "confirmed" for smoother flow.

Line25: "could result in an increase or decrease of microbial biomass parameters" change to “could result in fluctuations in microbial biomass parameters”

Line 27 : “dramatically increased microbial biomass” change to “substantially increased microbial biomass”

Line 30: “Supposedly mixed zones” sounds speculative. Clarify whether these zones are known or hypothesized.

Line 32–33: This sentence feels too loaded and would benefit from being split.

Line 34: “drinking water is a product prone to microbial deterioration during storage” change to “stored drinking water is highly susceptible to microbial deterioration”

Introduction

Lines 37–38: “In contrast to many other countries, drinking water in the Netherlands is distributed without a disinfectant residual.”; Lines 47–48: “Due to the high quality of drinking water and absence of odour/taste issues related to a disinfectant residual…”. These two statements could be merged or simplified to reduce redundancy.

Line 15: “premise plumbing systems”; Line 38 and onward: “premises plumbing system” (used multiple times). Standardize to either "premise plumbing system" or "premises plumbing system" throughout for consistency.

Lines 48–52: “Due to the high quality of drinking water and absence of odour/taste issues related to a disinfectant residual, it is common practice in the Netherlands that consumers refill their containers (i.e. bottles, water cookers, coffee reservoirs, etcetera) with drinking water from the tap. This offers both environmental and costs advantages…” This sentence could be broken into smaller units to improve readability and highlight the key message.

Line 47–48: “Due to the high quality of drinking water and absence of odour/taste issues related to a disinfectant residual…”. Consider reframing this as an observed behavior rather than a causal inference unless backed by behavioral data.

Lines 73–79: “Drinking water quality in the Netherlands is routinely monitored… heterotrophic plate counts at 22°C (HPC), Aeromonas plate counts and Legionella plate counts… ATP, AOC, biomass production potential, cell counts…”. This technical detail could be summarized more generally here and elaborated in the Methods section.

Line 54: “active microbial biomass”; Line 57: “active biomass (ATP)”; Line 67: “active biomass (ATP)”. Use consistent terminology and clarify what "active biomass" refers to throughout.

Lines 80–81:“This implies that the effect of drinking water storage on the microbial water quality might differ between drinking water treatment plants.” Consider citing specific studies or evidence to substantiate this hypothesis.

Lines 89–94:“…a ‘new dawn’ of citizen science… One of the issues involving citizen scientists… is whether they can provide reliable microbial sampling…”. More explanation on why citizen scientists are beneficial (e.g., scalability, trust-building, spatial coverage) would strengthen this section.

Between lines 44–47 and 48–54:There is a jump from technical and aesthetic complaints to consumer storage practices. Add a transition to clarify how consumer behavior is a logical next step in the narrative.

Materials and Method

Lines 108–114 can be improved by avoiding repetition of the full treatment chains. Instead, emphasize the key differences between the two plants: Plant A supplies the western part of Amsterdam and includes an additional dune infiltration step, while Plant B supplies the eastern part and omits this step. The dune infiltration in Plant A may enhance microbial stability and influence the bacterial community composition of the treated water.

Line 120: The study relies on participants sampling water after overnight stagnation, but it’s not clear if standardized instructions were given (e.g., stagnation duration, handling). Variability here may affect microbial load and comparability.

Lines 125-126: The phrase "the other 13 citizen scientists choose not to sample drinking water" is a bit misleading. It sounds like they didn’t participate at all. The sentence "non-drinking water samples were omitted" is vague. What does "non-drinking water" mean in this context? The logic flow between who did what and what was analyzed could be more explicit.

Lines 137–141: Using citizen scientists to conduct bacterial and fungal plate counts on dipslides is practical but lacks precision and introduces variability. Room temperature incubation is not standardized, and subjective CFU counting by untrained users is prone to bias. In addition, the text lacks info on whether participants were trained, how they handled aseptic techniques, or if contamination risks were managed.

Lines 165–179: are thorough, but it would be good to specify: final read depth per sample, percentage of reads removed during filtering steps, normalization method before downstream analysis.

The PERMANOVA assumptions (e.g., homogeneity of dispersion) are not discussed. State whether tests like PERMDISP were conducted to verify validity of PERMANOVA results.

Results

Lines 199 & 202: The use of “< 1.0 ng ATP/l” without defining the detection limit or measurement threshold is unclear. Explicitly state the detection limit of the ATP assay and whether "< 1.0" reflects below detection or censoring.

Lines 197–198 and 200–201: The repeated phrasing “flushed drinking water samples taken by the citizen scientists whose houses were supplied with drinking water from the distribution system of plant A/B” is overly long and redundant. Replace with concise alternatives such as “citizen samples from Plant A’s distribution zone.”

Although averages are given for ATP levels from Plant A (1.3 ± 0.8 ng/l) and Plant B (2.1 ± 1.0 ng/l), no statistical test is presented to assess whether the difference is significant. Include a statistical comparison (e.g., t-test or Mann-Whitney U) to support any claims of difference between distribution zones.

The section is purely descriptive; it doesn’t comment on what the difference in ATP levels may imply in terms of microbial stability, water age, or plant performance. Briefly interpret the implications—e.g., “This may suggest higher biological stability of water from Plant A.”

Line 196: The opening sentence states the ATP concentration “in the treated water” but doesn’t clarify whether these values were measured directly at the treatment plants, when, or under what conditions. Clarify if these measurements were made at the plant outlets and under comparable temporal conditions (e.g., also in June 2016).

Lines 311–320 : The observation of dramatic shifts in bacterial community composition in stored water is well reported. However, the term “dramatic” (line 315) is qualitative and subjective. Use more specific metrics (e.g., % shift in community dissimilarity, diversity indices) to support this claim.

Line 312–314: The association between decreased unclassified taxa and increased relative abundance of Proteobacteria and Actinobacteria is informative, but the biological implications of these shifts are not discussed. Add interpretation: Do these taxa reflect regrowth of environmental bacteria, potential health risks, or biofilm formers?

Lines 327–338 : The use of PCoA and Bray-Curtis dissimilarity is methodologically appropriate. However, the visual summary ("scattered across the PCoA plot") in line 334 should be quantified, e.g., via centroid distances or dispersion measures.

Line 336–338: The claim that changes in community composition are “specific for each household” is plausible, but should be supported by inter-household variance measures or discussion of household-specific factors (e.g., container type, storage time, temperature).

Lines 339–357 : The degree of change due to stagnation vs. distribution is stated, but the magnitude of community dissimilarity between these steps (e.g., PERMANOVA R² values) is not provided. Include statistical effect sizes to support claims of biological relevance.

Lines 345–349: The conclusion that community composition remained stable during distribution is valid but should be qualified by the limitations of 16S rRNA profiling (e.g., resolution to genus/species level, potential under-detection of low-abundance shifts).

Line 384–386: Instead of repeating “in containers from different locations,” streamline to “in multiple households.”

Lines 391–396: The distinction between growth during stagnation and storage is key but could be better highlighted by comparing fold changes or delta relative abundances between time points.

It would help to contextualize the potential ecological or health roles of these dominant OTUs, especially genera like Pseudomonas, Acinetobacter, and Enhydrobacter.

Lines 418–421: The use of statistical testing (ANOVA with Bonferroni) is appropriate and strengthens the conclusion. However, the effect size or p-values should be mentioned explicitly.

Line 423: The term “microbial water quality was slightly better” is vague and needs a clearer definition—was this based on ATP only? Or all parameters?

Lines 426–430: The conclusion that the mixing zone is primarily fed by Plant A is reasonable based on ATP/cell count comparisons, but hydraulic data or modeling would greatly strengthen this inference.

Line 436–440: Class-level comparisons are mentioned, but the ecological or functional relevance of these shifts is not discussed. For example, is a lower abundance of β-Proteobacteria associated with lower nutrient availability?

Line 442–448: The result that the mixing zone samples resemble Plant A samples is convincing, but again, would benefit from quantitative support (e.g., % shared OTUs, similarity indices).

Discussion

Line 455–456: The sentence presents citizen science as a valid approach but lacks a critical view of its limitations (e.g., sampling errors, lack of lab conditions). Include a more balanced assessment by addressing both the benefits and limitations of using citizen scientists for microbiological sampling.

467–470: Clarify whether variability was due to sampling, analytical technique, or participant handling.

471–473: Add information on whether instruction quality was evaluated (e.g., quiz, feedback), or how citizen scientists' sampling performance was validated pre-study.

475–477: The claim that no aberrant results were observed is strong but not supported by statistical evidence.

478–481 The conclusion that citizen scientists can "reliably sample and analyze" is quite broad given that only basic microbiological parameters were assessed. Rephrase to limit the scope of reliability to the tested parameters and context, and emphasize the need for caution in generalizing these findings.

499–501: Improve clarity by first acknowledging that some parameters may appear stable while others show growth, and distinguish between general vs. specific microbial indicators.

508–513: The critique of using ATP and flow cytometry as indicators is valid, but lacks depth regarding what alternatives might be feasible. e.g., qPCR, next-gen sequencing, biosensors.

533–537: explore potential confounders like plumbing material, usage patterns, or temperature.

539–547 Reference studies quantifying copper's antimicrobial effect in situ to support the claim.

Reviewer #2: I have attached an edited Word version of the manuscript with grammatical edits and comments. The authors should review again for grammar and consistency in language. The paper has implications for the field of public health. Public water systems may be in compliance with regard to bacterial contaminants; however, water may become contaminated with bacteria during the distribution system or in containers used to store water. Water containment systems (bottled water) are also at risk of microbial growth. The authors involved citizen scientists in the process of collecting water and conducting some lab analyses, which included bacterial and fungal plate counts.

• The authors should consider expanding their methods to clearly define how citizen scientists were recruited and trained, what their role was in the project (for example, in addition to collecting water samples from the tap, it seems that testing water bottles for microbes was the citizen scientists’ idea, so their role was to reliably collect samples, assist in bacterial and fungal plate counts, and pose new questions about sources and locations of drinking water contamination), and when and how they were included in laboratory analyses.

• Are any of the citizen scientists authors?

• The description of citizen science outcomes in the Discussion should be in Results.

• How does this paper build on and/or relate to reference 37? Are there any citizen scientists who overlap between these two endeavors?

Overall, this is a thorough paper. It is somewhat challenging to determine whether it primarily focuses on citizen science or the contamination of drinking water through distribution systems assisted by citizen scientists. The authors should review each section of the manuscript to highlight which of their approaches was central to the questions addressed in this study.

**Do you want your identity to be public for this peer review?** For information about this choice, including consent withdrawal, please see our Privacy Policy

Reviewer #1: **Yes: ** Suhyb Salama

Reviewer #2: No

---

## [Author Response · Author response to Decision Letter 1]

25 Jul 2025

Reviewer #1: The paper addresses an important and timely topic: the vulnerability of non-chlorinated drinking water to microbial deterioration during distribution, stagnation, and storage, using a citizen science approach. This is highly relevant for public health monitoring and community engagement, particularly in regions like the Netherlands where drinking water is distributed without residual disinfectants. However, the manuscript, while methodologically sound in many respects, suffers from a few limitations in clarity and structure. These include inconsistencies in phrasing, unclear sample classification, and a lack of discussion around potential variability introduced by citizen-led procedures. The limitations of dipslide-based microbial counts, the need for standardization in sampling and incubation, and clarification of the analytical pipeline, should be addressed to strengthen the scientific rigor and readability of the manuscript. The conclusions should be more cautious given the exploratory nature and dataset limitations.

We thank the reviewer for the extensive review of our manuscript and the comments and suggestions made, which helped us to improve the manuscript, especially on the statistical analysis part. The points raised above are also given in the specific comments of the reviewer and below is our point-to-point response to each of these comments. Please note that the reference to line numbers in the comments of the reviewer refers to the original submission, whereas the reference to line numbers in our reply refers to the revised manuscript with track changes.

Abstract

1. Lines 15–17: The opening sentence is overly long and could be made more digestible.

The sentence has been shortened (Lines 15-16).

2. Lines 20–21: The phrase “prokaryotic cell counts using flow cytometry, ATP concentrations and the prokaryotic community composition was determined” contains a subject-verb agreement error and could be clearer.

The subject-verb agreement was corrected, and the sentence was made clearer (Lines 20-22).

3. Line 23: Replace "demonstrated" with "showed" or "confirmed" for smoother flow.

We replaced demonstrated with showed.

4. Line25: "could result in an increase or decrease of microbial biomass parameters" change to “could result in fluctuations in microbial biomass parameters”

Changed accordingly.

5. Line 27 : “dramatically increased microbial biomass” change to “substantially increased microbial biomass”

Changed accordingly, also at other places in the manuscript.

6. Line 30: “Supposedly mixed zones” sounds speculative. Clarify whether these zones are known or hypothesized.

We understand the confusion. Supposedly was used, because the drinking water companies sometimes do not exactly know where drinking water types are mixed in their distribution system. However, you can also read it as being speculative. For clearance, we removed the word supposedly from the sentence.

7. Line 32–33: This sentence feels too loaded and would benefit from being split.

This sentence has been split and rephrased, also in relation to a comment made by Reviewer 2 (Lines 31-34).

8. Line 34: “drinking water is a product prone to microbial deterioration during storage” change to “stored drinking water is highly susceptible to microbial deterioration”

Changed accordingly.

Introduction

9. Lines 37–38: “In contrast to many other countries, drinking water in the Netherlands is distributed without a disinfectant residual.”; Lines 47–48: “Due to the high quality of drinking water and absence of odour/taste issues related to a disinfectant residual…”. These two statements could be merged or simplified to reduce redundancy.

We have rephrased this part to eliminate the redundancy (lines 40-41).

10. Line 15: “premise plumbing systems”; Line 38 and onward: “premises plumbing system” (used multiple times). Standardize to either "premise plumbing system" or "premises plumbing system" throughout for consistency.

Changed consistently in premises plumbing system(s).

11. Lines 48–52: “Due to the high quality of drinking water and absence of odour/taste issues related to a disinfectant residual, it is common practice in the Netherlands that consumers refill their containers (i.e. bottles, water cookers, coffee reservoirs, etcetera) with drinking water from the tap. This offers both environmental and costs advantages…” This sentence could be broken into smaller units to improve readability and highlight the key message.

Sentence has been shortened (Lines 52-54).

12. Line 47–48: “Due to the high quality of drinking water and absence of odour/taste issues related to a disinfectant residual…”. Consider reframing this as an observed behavior rather than a causal inference unless backed by behavioral data.

We have rephrased this part in relation to comment 9, which also resolves this comment.

13. Lines 73–79: “Drinking water quality in the Netherlands is routinely monitored… heterotrophic plate counts at 22°C (HPC), Aeromonas plate counts and Legionella plate counts… ATP, AOC, biomass production potential, cell counts…”. This technical detail could be summarized more generally here and elaborated in the Methods section.

This part does not describe the analysis we did in our study, but it shows what parameters drinking water companies in The Netherlands routinely monitor to determine the microbiological drinking water quality. Since not all these parameters were measured in our study, it does not fit into the Materials and Methods section that only describes the methods used in our study. As such, we have kept this sentence as it was.

14. Line 54: “active microbial biomass”; Line 57: “active biomass (ATP)”; Line 67: “active biomass (ATP)”. Use consistent terminology and clarify what "active biomass" refers to throughout.

We have defined that ATP is used as a measure for active biomass on lines 60-61. We also used active biomass as the consistent terminology throughout the revised manuscript.

15. Lines 80–81:“This implies that the effect of drinking water storage on the microbial water quality might differ between drinking water treatment plants.” Consider citing specific studies or evidence to substantiate this hypothesis.

This hypothesis is derived from the observation that microbiological water quality parameters, biological stability parameters and bacterial community composition differs between production locations, which we think is clearly described in the preceding sentence. We feel that the word ‘This’ in the sentence clearly points to these differences described directly before this sentence. As such, we have kept this sentence as it is.

16. Lines 89–94:“…a ‘new dawn’ of citizen science… One of the issues involving citizen scientists… is whether they can provide reliable microbial sampling…”. More explanation on why citizen scientists are beneficial (e.g., scalability, trust-building, spatial coverage) would strengthen this section.

It is beyond the scope of our manuscript to write a (show) review on potential and limitations of citizen science in the Introduction section, but such aspects are described in the references provided. Therefore, we added that these aspects can be found in the references provided (Lines 97-98).

17. Between lines 44–47 and 48–54:There is a jump from technical and aesthetic complaints to consumer storage practices. Add a transition to clarify how consumer behavior is a logical next step in the narrative.

See our reply to comment 9 and 12.

Materials and Method

18. Lines 108–114 can be improved by avoiding repetition of the full treatment chains. Instead, emphasize the key differences between the two plants: Plant A supplies the western part of Amsterdam and includes an additional dune infiltration step, while Plant B supplies the eastern part and omits this step. The dune infiltration in Plant A may enhance microbial stability and influence the bacterial community composition of the treated water.

Changed accordingly.

19. Line 120: The study relies on participants sampling water after overnight stagnation, but it’s not clear if standardized instructions were given (e.g., stagnation duration, handling). Variability here may affect microbial load and comparability.

Participants watched an instruction video on how samples have to be taken during a meeting with them and this video remained accessible through a website that was active during the runtime of the project. We added this information on lines 132-135.

20. Lines 125-126: The phrase "the other 13 citizen scientists choose not to sample drinking water" is a bit misleading. It sounds like they didn’t participate at all. The sentence "non-drinking water samples were omitted" is vague. What does "non-drinking water" mean in this context? The logic flow between who did what and what was analyzed could be more explicit.

We have changed the text to make this aspect more explicit on lines 143-145.

21. Lines 137–141: Using citizen scientists to conduct bacterial and fungal plate counts on dipslides is practical but lacks precision and introduces variability. Room temperature incubation is not standardized, and subjective CFU counting by untrained users is prone to bias. In addition, the text lacks info on whether participants were trained, how they handled aseptic techniques, or if contamination risks were managed.

One of the aims was to determine whether citizen scientists can be used to reliable sample drinking water for microbiology. The dip slides used were chosen because they have been developed for testing by non-trained personnel and the manual instructions of the methods have been followed by the participants (as stated on line 159), limiting variability and bias. The limited variability and bias can be seen by the results from the drinking water samples, which showed no aberrant colony counts, ATP concentrations, cell counts or community composition between samples taken by the different citizen scientists nor with the samples that were taken and analyzed by trained personnel (as described in the Results section). Furthermore, keeping citizen scientists motivated requires more than just sampling drinking water. Therefore, the cultivation on dip slides was a crucial part for the citizen scientist’s involvement. One of the remarkable outcomes is that our study indicates that, when instructed properly, citizen scientists can reliably sample and analyze drinking water (as we concluded on lines 688-690).

22. Lines 165–179: are thorough, but it would be good to specify: final read depth per sample, percentage of reads removed during filtering steps, normalization method before downstream analysis.

The final read depth has been added (Lines 207-208). The percentage of reads removed during filtering steps has not been documented and cannot be given. The normalization method has been described on lines 213-219.

23. The PERMANOVA assumptions (e.g., homogeneity of dispersion) are not discussed. State whether tests like PERMDISP were conducted to verify validity of PERMANOVA results.

We thank the reviewer for pointing this out and based on this comment we have performed a PERMDISP analysis. The results of the PERMDISP have been added to the results section where the PERMANOVA results are presented (lines 387-390, 402-403 and 415-417).

Results

24. Lines 199 & 202: The use of “< 1.0 ng ATP/l” without defining the detection limit or measurement threshold is unclear. Explicitly state the detection limit of the ATP assay and whether "< 1.0" reflects below detection or censoring.

The detection limit of the ATP assay was added to line 167.

25. Lines 197–198 and 200–201: The repeated phrasing “flushed drinking water samples taken by the citizen scientists whose houses were supplied with drinking water from the distribution system of plant A/B” is overly long and redundant. Replace with concise alternatives such as “citizen samples from Plant A’s distribution zone.”

Changed accordingly

26. Although averages are given for ATP levels from Plant A (1.3 ± 0.8 ng/l) and Plant B (2.1 ± 1.0 ng/l), no statistical test is presented to assess whether the difference is significant. Include a statistical comparison (e.g., t-test or Mann-Whitney U) to support any claims of difference between distribution zones.

We believe that the reviewer refers here to the section with subheading ‘Influence of distribution, overnight stagnation and storage of drinking water’. We refer to Table 1 in this section to show differences in ATP concentrations between samples taken at the treatment plant, directly and after flushing at the tap, and after storage. However, Table 1 also contains information on the differences between Plant A and Plant B. However, that comparison is described in the section with subheading “Influence of treatment plant”. There, we also presented the results from the statistical test on differences in ATP-concentrations between Plant A samples, Plant B samples and samples that received either Plant A or Plant B. Since three groups are compared an ANOVA test with Bonferroni posthoc test was done. We refer the reviewer to lines 488-492 where this part is described.

27. The section is purely descriptive; it doesn’t comment on what the difference in ATP levels may imply in terms of microbial stability, water age, or plant performance. Briefly interpret the implications—e.g., “This may suggest higher biological stability of water from Plant A.”

In our study, ATP was only determined as a measure of active biomass. Our previous research showed that ATP is not a reliable indicator for (micro)biological stability, plant performance or water age (van der Wielen & van der Kooij, 2010, Water Res 2010 Vol. 44 Issue 17 Pages 4860-4867 and van der Wielen et al. 2023, Science of The Total Environment 2023 Vol. 871 Pages 161930). However, ATP can be used as an indication for regrowth, maintenance or death. This was already partly included in the original manuscript where we stated that: “At 19 of these 25 locations the ATP increased with more than 20%, indicating microbial growth in the premises plumbing system during overnight stagnation at these 19 locations.”, and “Practically all locations showed more than 20% higher ATP concentrations or cell numbers in stored drinking water than in flushed drinking water samples from the tap, demonstrating microbial growth occurs during drinking water storage in different containers.” On other locations, we included additional text to shortly describe the implications of ATP-changes (Lines 276-277, 290-293, 487-488).

28. Line 196: The opening sentence states the ATP concentration “in the treated water” but doesn’t clarify whether these values were measured directly at the treatment plants, when, or under what conditions. Clarify if these measurements were made at the plant outlets and under comparable temporal conditions (e.g., also in June 2016).

This information is provided in the Materials and Methods section on lines 147-148.

29. Lines 311–320: The observation of dramatic shifts in bacterial community composition in stored water is well reported. However, the term “dramatic” (line 315) is qualitative and subjective. Use more specific metrics (e.g., % shift in community dissimilarity, diversity indices) to support this claim.

A percentage shift in community composition is difficult to present here as the percentage shift is not the same for all samples and differs quite a bit between each location where stored drinking water was compared with drinking water from the tap. Therefore, to be more neutral, we omitted the term dramatic in this sentence.

30. Line 312–314: The association between decreased unclassified taxa and increased relative abundance of Proteobacteria and Actinobacteria is informative, but the biological implications of these shifts are not discussed. Add interpretation: Do these taxa reflect regrowth of environmental bacteria, potential health risks, or biofilm formers?

The phylum Proteobacteria or the phylum Actinobacteria contain species that differ in their metabolic capacities. For instance, the phylum Proteobacteria contain species that can oxidize ammonia, reduce sulfate, aerobically or anaerobically oxidize organic compounds, can be pathogenic, non-pathogenic or even beneficial. Consequently, reliable assigning function to a

---

## [Decision Letter · Decision Letter 1]

25 Sep 2025

Dear Dr. van der Wielen,

Thank you for submitting your manuscript to PLOS ONE. After careful consideration, we feel that it has merit but does not fully meet PLOS ONE’s publication criteria as it currently stands. Therefore, we invite you to submit a revised version of the manuscript that addresses the points raised during the review process.

**The reviewers have made their comments on the revised manuscript and found that most of the prior concerns from the original submission have been addressed. Please see the notes from reviewer #2 for additional improvements and edits that should be included. **

We look forward to receiving your revised manuscript.

Kind regards,

Theodore Raymond Muth

Academic Editor

PLOS ONE

**Journal Requirements:**

Reviewers' comments:

Reviewer's Responses to Questions

**Comments to the Author**

Reviewer #1: All comments have been addressed

Reviewer #2: (No Response)

2. Is the manuscript technically sound, and do the data support the conclusions?

Reviewer #1: Yes

Reviewer #2: Yes

3. Has the statistical analysis been performed appropriately and rigorously?

Reviewer #1: Yes

Reviewer #2: Yes

4. Have the authors made all data underlying the findings in their manuscript fully available?

Reviewer #1: Yes

Reviewer #2: Yes

5. Is the manuscript presented in an intelligible fashion and written in standard English?

Reviewer #1: Yes

Reviewer #2: Yes

**Reviewer #1: ** The authors have used this revision cycle seriously to significantly improve the manuscript and enhance its scientific soundness to a very good level. All comments have been properly addressed.

**Reviewer #2: ** The edits contribute significantly to the readability and understandability of the manuscript. I am not sure that you can insert references the way you did, adding an "a" rather than re-ordering, but I leave it to the editors to decide how to deal with references.

In the clean version, I see some typos and need for grammatical changes:

Line 119: Do you mean citizen scientists obtained water samples from these districts? So, they were not just testing their homes, but also collected other samples as well? I am not sure how their homes would be supplied by both districts A and B.

Line 132: you mean “chose” not “choose”.

Line 150: houses of citizen scientists

Line 163-164: “flushed and direct drinking water and between direct and stored drinking water samples

Line 174: Solution BF1 from the kit

Line 181-182: perhaps the “Illumina MiSeq 16S Metagenomic sequencing 182 library preparation protocol” should be cited in references rather than as a url in the text of the manuscript.

Line 189: same comment for the MiSeq standard operating procedure

Line 220: The results being reported here aren’t about the citizen scientists but about their samples. The samples revealed ATP concentrations…that should be the subheading here…” ATP Concentrations from Samples Supplied by Citizen Scientists”. Or even just ….”ATP Concentrations”.

Line 274: remove the word “that” to read “indicating active biomass decay”

Line 300: Replace “practically all” with “most”

Line 303: Replace “had been” with “were”

Line 310-311: Seems odd to describe the shift toward a community of bacteria “capable of growing on an agar medium”. What exactly was the makeup of the bacterial community that was growing on an agar medium? Did the medium select for a particular type? If so, state that.

Line 428: Replace “belonged” with “belong”

Line 487: Replace “Proteobaceria” with “Proteobacteria.”

Line 505: Replace “unexperienced” with “inexperienced”

Line 543: a premise’s….this is a recurring issue…I think it should be the possessive in each case throughout the manuscript.

Line 550-553: Rather than say that it would be a misconception to infer from the findings that regrowth of bacteria in distribution systems is negligible…Simply state that “although we did not identify significant regrowth of bacteria in distribution systems, other studies have verified that this can occur”…then go on to reference studies with a different outcome from yours.

Line 553: I don’t know what en Legionella means.

Line 590: cite “previous studies”

These are all easily addressed.

**Do you want your identity to be public for this peer review?** For information about this choice, including consent withdrawal, please see our Privacy Policy

Reviewer #1: **Yes: ** Suhyb Salama

Reviewer #2: No

---

## [Author Response · Author response to Decision Letter 2]

4 Oct 2025

Dear Editor,

We thank the reviewers for their positive judgement of our revised submission and that they noticed that we addressed most of their prior concerns. From the review reports we understood that only reviewer 2 has made some suggestions from improvements and we thank the reviewer for pointing out these improvements. We address these comments below. Please note that the reference to line numbers in the comments of the reviewer refers to the original submission, whereas the reference to line numbers in our reply refers to the revised manuscript with track changes.

The edits contribute significantly to the readability and understandability of the manuscript. I am not sure that you can insert references the way you did, adding an "a" rather than re-ordering, but I leave it to the editors to decide how to deal with references.

Response: We have added an ‘a’ rather than reordering the whole reference list in other publications, although these publications were not published in PLOS ONE. We will wait for the editors how they decide on this.

In clean version, I see some typos and need for grammatical changes:

Line 119: Do you mean citizen scientists obtained water samples from these districts? So, they were not just testing their homes, but also collected other samples as well? I am not sure how their homes would be supplied by both districts A and B.

Response: The 8 houses where the reviewer refers to received either drinking water from plant A or plant B depending on the time of the day. We have clarified this point on lines 119 to 121.

Line 132: you mean “chose” not “choose”.

Changed accordingly.

Line 150: houses of citizen scientists

Changed accordingly.

Line 163-164: “flushed and direct drinking water and between direct and stored drinking water samples

Changed accordingly.

Line 174: Solution BF1 from the kit

Changed accordingly.

Line 181-182: perhaps the “Illumina MiSeq 16S Metagenomic sequencing 182 library preparation protocol” should be cited in references rather than as a url in the text of the manuscript.

Response: In our previous publications we have always referred to this document in the manuscript text and not in the reference list, as these documents do not have a specified author. Therefore, we have kept the url in the text here.

Line 189: same comment for the MiSeq standard operating procedure.

Response: In our previous publications we have always referred to this document in the manuscript text and not in the reference list, as these documents do not have a specified author. Therefore, we have kept the url in the text here.

Line 220: The results being reported here aren’t about the citizen scientists but about their samples. The samples revealed ATP concentrations…that should be the subheading here…” ATP Concentrations from Samples Supplied by Citizen Scientists”. Or even just ….”ATP Concentrations”.

Response: In the first part of the results section (i.e. under the first heading in the results section where the reviewer points to) we want to show the sampling performance of the citizen scientists. This section is not about the ATP concentrations we measured as ATP results are also presented under the other headings in the Results section. Therefore, renaming the heading to ATP-concentrations doesn’t seem to cover the results presented here. Therefore, we renamed the heading: Sampling performance by the citizen scientists (line 220).

Line 274: remove the word “that” to read “indicating active biomass decay”

Changed accordingly.

Line 300: Replace “practically all” with “most”

Changed accordingly.

Line 303: Replace “had been” with “were”

Changed accordingly.

Line 310-311: Seems odd to describe the shift toward a community of bacteria “capable of growing on an agar medium”. What exactly was the makeup of the bacterial community that was growing on an agar medium? Did the medium select for a particular type? If so, state that.

Response: As described in Materials and Methods we used agar dip slides that detect bacteria. To be more precise we changed the sentence into ‘This result is an indication that not only the biomass increased during drinking water storage but that the community also shifted towards one harboring more bacteria capable of growing on the agar dip slides for bacteria used in our study’ (lines 308-311).

Line 428: Replace “belonged” with “belong”

Changed accordingly.

Line 487: Replace “Proteobaceria” with “Proteobacteria.”

Changed accordingly.

Line 505: Replace “unexperienced” with “inexperienced”

Changed accordingly.

Line 543: a premise’s….this is a recurring issue…I think it should be the possessive in each case throughout the manuscript.

Response: According to the English dictionary premise refers to an assertion or proposition, whereas premises refers to the land and buildings that are part of a property. Since the plumbing systems refers to physical property in a building, we believe premises plumbing is correct English. So, we have kept premises throughout the manuscript.

Line 550-553: Rather than say that it would be a misconception to infer from the findings that regrowth of bacteria in distribution systems is negligible…Simply state that “although we did not identify significant regrowth of bacteria in distribution systems, other studies have verified that this can occur”…then go on to reference studies with a different outcome from yours.

Response: We have not reliably determined regrowth in our study, because we already know from previous studies that more parameters need to be determined to identify regrowth. We agree with the reviewer that the word misconception might be too strong and, therefore, we have removed that part from the manuscript (lines 550-551).

Line 553: I don’t know what en Legionella means.

Response: We apologize for this. ‘en’ is a Dutch word meaning and. We changed that in the revised manuscript.

Line 590: cite “previous studies”

Changed accordingly.

---

## [Editor Report · Decision Letter 2]

7 Oct 2025

Microbial drinking water quality deterioration during distribution and household usage, determined together with citizen scientists.

PONE-D-25-20999R2

Dear Dr. van der Wielen,

We’re pleased to inform you that your manuscript has been judged scientifically suitable for publication and will be formally accepted for publication once it meets all outstanding technical requirements.

Kind regards,

Theodore Raymond Muth

Academic Editor

PLOS ONE
---

## [Editor Report · Acceptance letter]

PONE-D-25-20999R2

PLOS ONE

Dear Dr. van der Wielen,

I'm pleased to inform you that your manuscript has been deemed suitable for publication in PLOS ONE. Congratulations! Your manuscript is now being handed over to our production team.

Kind regards,

on behalf of

Dr. Theodore Raymond Muth

Academic Editor

PLOS ONE